# Derived Neutrophil-Lymphocyte Ratio and C-Reactive Protein as Prognostic Factors for Early-Stage Non-Small Cell Lung Cancer Treated with Stereotactic Body Radiation Therapy

**DOI:** 10.3390/diagnostics13020313

**Published:** 2023-01-14

**Authors:** Baiqiang Dong, Xuan Zhu, Runzhe Chen, Qing Wu, Jia’nan Jin, Lin Wang, Yujin Xu, Ming Chen

**Affiliations:** 1Department of Radiation Oncology, Sun Yat-sen University Cancer Center, State Key Laboratory of Oncology in South China, Collaborative Innovation Center for Cancer Medicine, Sun Yat-sen University, Guangzhou 510000, China; 2Department of Thoracic Radiotherapy, The Cancer Hospital of the University of Chinese Academy of Sciences (Zhejiang Cancer Hospital), Institute of Basic Medicine and Cancer (IBMC), Chinese Academy of Sciences, Zhejiang Key Laboratory of Radiation Oncology, Hangzhou 310024, China; 3Department of Radiation Oncology, the First Affiliated Hospital, College of Medicine, Zhejiang University, Hangzhou 310027, China; 4The Second Clinical Medical College of Zhejiang Chinese Medicine University, Hangzhou 310053, China; 5School of Medicine, Shaoxing University, Shaoxing 312010, China

**Keywords:** inflammation, C-reactive protein, derived neutrophil-lymphocyte ratio, stereotactic body radiotherapy, non-small cell lung cancer

## Abstract

*Objectives*: To explore the relationship between peripheral blood inflammation parameters and overall survival (OS) and progression-free survival (PFS) of early-stage non-small cell lung cancer patients who underwent stereotactic body radiotherapy (SBRT). *Patients and methods*: In this study, eligible patients treated with SBRT from 2013 to 2018, and both serum complete blood count and blood biochemical results were available prior to (within 60 days) radiotherapy were included. *Results*: A review of hospital registries identified 148 patients, and the 5-year OS and PFS of the entire cohort were 69.8% and 65.6%, respectively, with the median follow-up time was 52.8 months. Multivariable analysis showed that derived neutrophil-lymphocyte ratio (dNLR) ≥1.4 and C-reactive protein (CRP) ≥2.9 were statistically and independently associated with worse OS (HR = 4.62, 95% CI 1.89–11.27, *p* = 0.001; HR = 2.92, 95% CI 1.49–5.70, *p* = 0.002, respectively). The 5-year OS for patients with dNLR below and equal to or above the 1.4 were 85.3% and 62.9% (*p* = 0.002), respectively, and 76.7% for the low CRP group versus 58.5% for the high CRP group (*p* = 0.030). Higher serum level of post-treatment CRP also independent parameters for inferior PFS (HR = 4.83, 95% CI 1.28–18.25, *p* = 0.020). *Conclusions*: Our results demonstrate that dNLR and CRP are associated with the outcomes of early-stage NSCLC patients treated with SBRT, which may assist in selecting optimal nursing care and therapeutic scheme for every individual.

## 1. Introduction

Non-small cell lung cancer (NSCLC) is the primary reason of cancer mortality globally, with approximately 16% of diagnosed cases at an early stage. Lobectomy with mediastinal lymph node dissection offer the best potential cure for patients with early-stage NSCLC. For individuals who were either medically inoperable or refused surgery, stereotactic body radiotherapy (SBRT) is a principal alternative treatment. As the use of SBRT continues to advance, it has also been explored and shown promising outcomes in potentially operable patients [1,2]. Nevertheless, in the group of inoperable patients for whom SBRT is commonly prescribed, the overall survival (OS) and progression-free survival (PFS) remain unsatisfactory [3,4,5]. Therefore, simple inexpensive-independent prognostic factors derived from readily available laboratory values would aid clinicians choose the optimal care and treatment strategy.

SBRT has been observed to yield greater antitumor effectiveness than would be predicted from standard radiobiologic modeling alone, possibly through superior engagement of the immune system, leading to enhanced antitumor immunity [6,7,8]. After the immune system stimulation, pro-inflammatory cytokine production by immune-related cells causes systemic inflammation. The correlation between diagnostic factors of systemic inflammation, including peripheral blood cells and lymphocyte proportions, as well as associated diseases’ prognoses in several forms of cancer, including NSCLC, has been investigated [9,10,11]. Nevertheless, the majority of these investigations are mainly focused on surgery, chemotherapy, and standard radiation, with scant attention paid to early-stage NSCLC patients who had SBRT.

A few previous studies have attempted to evaluate the prognostic role of peripheral blood cells and lymphocyte ratios, such as neutrophil-lymphocyte ratio (NLR), monocyte-lymphocyte ratio (MLR), and platelet-lymphocyte ratio (PLR), et al., in the setting of lung SBRT [12,13,14,15,16]. The results of these studies, however, appear inconsistent, which may attribute to the limited median follow-up time (range from 13.4 to 29.5 months) or smaller sample size (range from 59 to 156 cases). Here, our research’s objective was to evaluate the prognostic significance of starting point obtained NLR (dNLR), MLR, PLR, and serum levels of albumin, lactate dehydrogenase (LDH), and C-reactive protein (CRP) in early-stage NSCLC patients administrated SBRT using a bigger sample size and a database with extensive follow-up. Furthermore, we assessed the role of post-treatment dNLR, serum albumin levels, and CRP in predicting clinical benefit, which had not been investigated before.

## 2. Patients and Methods

### 2.1. Patient Selection

This research was authorized from the Institutional Review Board of Zhejiang Cancer Hospital and due to the retrospective nature of the research, it was not necessary to obtain the participants’ written consent. Patients who had clinically confirmed early-stage NSCLC (T1-2N0M0) with no prior history of malignant tumors, treatment with SBRT to a biologically effective dose (BED_10_) ≥ 100 Gy, and availability of both serum complete blood count (CBC) and blood biochemical (BBC) results prior to (within 60 days) radiotherapy were selected for further analysis. The exclusion criteria for the involved subjects were as follow; had local-regional recurrence disease, small-cell histologic characteristics, metastatic lung cancer, who had not received SBRT alone, or who were undergoing radiotherapy for palliative purposes. The anonymous laboratory results were taken from the digital patient database. Additionally, demographical, clinical, and pathological data were collected. In the investigation of post-treatment hematological and serum metrics, a subgroup of patients who provided CBC and BBC with differential following 60 days of SBRT termination was analyzed.

### 2.2. Staging and Treatment Procedures

Before SBRT, a computed tomography (CT) scan of the chest and upper abdomen, a magnetic resonance imaging (MRI) scan of the brain, a bone scan for staging, and blood samples were taken. Clinically proven lung cancer was defined as an initial suspected mass, or part-solid or ground-glass opaque nodules with speculative or sleek borders on CT images that lasted for 3 months or more and grew along its longitudinal axis. Endobronchial ultrasonography or mediastinoscopy was conducted at the judgment of the clinician for patients with ambiguous correlations across radiological findings. When a biopsy was judged clinically unsafe or the patient refused to undergo the procedure, positron emission tomography/computed tomography (PET/CT) was declared required for diagnosing all patients. The eighth version of the American Joint Committee on Cancer (AJCC) standards were used to grade the tumors [17].

For treatment planning, all patients had a free-breathing 4-dimensional CT (4DCT) simulation scanning. The maximum intensity projection (MIP) data were derived by applying the highest density value to every pixel across all 10 phases of 4DCT imaging. The depiction of tumor size has been described in our prior investigations [18,19]. On the MIP datasets, the internal target volume (ITV) was sculpted. The planned target volume (PTV) was defined as the sum of the ITV and an isotropic expansion of 5 mm. Organs at risk were identified, and the maximum point dose restrictions for surrounding structures complied with the Radiation Therapy Oncology Group (RTOG) 0236 recommendations for peripheral tumors and the recognized limits for centralized tumors [20]. In SBRT, the prescribed dosage was necessary to treat 95% of the PTV, and radiotherapy was administered daily with a routine image guide (cone beam CT) to validate the site of the target.

### 2.3. Follow-Up and Statistical Analysis

Follow-up consisted of a physical checkup and CT scanning every three months for the initial three years, followed by a CT scanning every six months to evaluate the therapy’s efficacy and toxicity. In cases of high probability of recurring, such as a rise in consolidation at the treated area or the formation of a new lung nodule or an enlarged lymph node, a PET/CT was recommended. If there was still doubt, a biopsy and/or a multidisciplinary tumor board discussions were conducted. For patients who underwent follow-up checks at other hospitals, information was obtained through telephone interviews either with the patient or with a relative.

The dNLR was calculated as neutrophil count/(leucocyte count-neutrophil count). Local regional recurrence (LRR) was identified as failure at the SBRT field site and/or regional nodes, such as the mediastinal and hilar basins. The term ‘distant metastasis’ (DM) refers to relapse outside of the treated lobe and regional lymph nodes. OS was measured from the time of the initial SBRT treatment till the day of death from any reason. PFS was estimated from the day of the initial SBRT treatment to advancement of cancer or death from any reason. Receiver operating characteristic (ROC) assessment was performed to find the appropriate cut-off value for each peripheral blood indicator that provides the highest reliability and validity. In accordance with these cutoff values, a Kaplan-Meier study of OS and PFS was conducted, and variations among every pair of groups were evaluated using the log-rank testing. Using a univariable Cox proportional hazard approach, the hazard ratio (HR) and 95% confidence interval (CI) were determined. In univariable analysis, criteria with *p* < 0.1 were chosen for multivariable analysis. All statistical tests were two-sided, and variables with *p* < 0.05 were judged to be statistically significant. SPSS version 24.0 (IBM Corp., Armonk, NY, USA) and GraphPad Prism 8.3.0 (GraphPad Software, San Diego, CA, USA) were used for all analyses.

## 3. Results

### 3.1. Patient Characteristics

Between January 2013 and August 2018, 148 eligible patients were enrolled for further analysis with male patients accounting for the majority (72%). The median age of the entire cohort was 76 years (range 47–89 years). In 48 patients (33%), histology was adenocarcinoma; in 27 (18%), it was squamous cell carcinoma; in 21 (14%), it was NSCLC, not otherwise specified; and the remaining patients (35%) refused to provide biopsies. A total of 150 patients (78%), involving those having no pathological affirmation, received PET/CT examination before SBRT. The tumor in 129 patients (87%) was clinical stage T1, and in 19 (13%), it was T2. The median total dosage, dose-per-fraction, and BED_10_ of SBRT were 50 Gy (range 50–70 Gy), 10.0 Gy (range 7.0–12.5 Gy), and 100 Gy (range 100–132), respectively.

Median neutrophil count at baseline was 3.5 × 10^9^/L (range 0.9–9.6 × 10^9^/L), median lymphocyte count was 1.5 × 10^9^/L (range 0.6–5.4 × 10^9^/L), median LDH was 186 U/L (range 109–354 U/L), and median CRP was 2.2 mg/L (range 0.0–89.5 mg/L; Appendix A). The median dNLR, MLR, and PLR of the cohort was 1.5 (range 0.3–7.7), 0.3 (range 0.0–1.2), and 128.2 (range 36.5–468.3), respectively. The baseline characteristics and the detailed information of the demographic were shown in Table 1.

### 3.2. Pre-Treatment dNLR and CRP as Prognostic Biomarkers

The median follow-up time was 52.8 months (range 3.9–110.4 months). During the last follow-up visit, 48 patients died, including 29 deaths caused by known tumor progression; Other causes of death included non-tumor factors such as heart failure, chronic obstructive pulmonary disease and accidental falls. Local regional recurrence occurred in 25 patients (eight biopsy-proven), and distant failures occurred in 41 patients; eight patients have developed second primary carcinomas, including second primary lung cancer (5 patients), melanoma, primary liver cancer and cervical cancer. The 3- and 5-year OS and PFS of the entire cohort were 83.5%, 69.8%, and 75.0%, 65.6%, respectively (Figure 1).

Based on ROC analysis, optimal cut-off values of dNLR, MLR, PMR, albumin, LDH and CRP were 1.4, 0.3, 128, 43, 240 and 2.9, respectively (Appendix A). On univariate cox regression analysis, we found a statistically significant association of pre-treatment dNLR ≥ 1.4 with higher overall mortality (HR = 3.38, 95% CI 1.51–7.58, *p* = 0.003). By contrast, similar significant trend has not been observed in high MLR group (*p* = 0.119) or high PLR group (*p* = 0.868). As for BBC, univariable analysis demonstrated that the pre-treatment serum level of albumin <43.0 (HR = 2.50, 95% CI 1.30–4.82, *p* = 0.006) and CRP ≥ 2.9 (HR = 1.85, 95% CI 1.05–3.27, *p* = 0.033) were statistically significant correlated with worse OS. No significant difference of OS was found between low and high LDH groups (*p* = 0.954).

Multivariable analysis showed that, only dNLR ≥ 1.4 and CRP ≥ 2.9 continued to be statistically and independently linked to worse OS (HR = 4.62, 95% CI 1.89–11.27, *p* = 0.001; HR = 2.92, 95% CI 1.49–5.70, *p* = 0.002, respectively) when accounting for age, gender, Eastern Cooperative Oncology Group performance status (ECOG PS), Charlson comorbidity index (CCI), pulmonary function, smoking status, tumor pathologic type, T-stage, and BED_10_ (Table 2). The 5-year OS for patients with dNLR below and equal to or above the 1.4 were 85.3% and 62.9%, respectively (*p* = 0.002; Figure 2A), and 76.7% for the low CRP group versus 58.5% for the high CRP group (*p* = 0.030; Figure 2B). There was no statistically significant association of pre-treatment dNLR ≥ 1.4 with LRR (*p* = 0.791), DM (*p* = 0.069) or PFS (*p* = 0.100) (Appendix A). However, CRP ≥ 2.9 remained independent significant parameters for inferior LRR (HR = 17.17; 95% CI 5.56–53.08; *p* < 0.001), DM (HR = 1.97; 95% CI 1.04–3.72; *p* = 0.038), and PFS (HR = 3.13; 95% CI 1.63–6.01; *p* = 0.001) (Appendix A).

### 3.3. Post-Treatment CRP as a Prognostic Biomarker

Considering the observation that the pre-treatment dNLR and CRP were significantly prognostic for early-stage NSCLC patients receiving SBRT, we investigated whether post-treatment metrics were also prognostic in the subgroup cohort. Notably, few patients (*n* = 46) had post-treatment CBC and BBC findings, most probably related to institutional surveillance practice patterns. We studied variations in peripheral blood indicators before and after treatment (Appendix A). The median post-treatment dNLR and CRP were 1.9 (range 0.7–5.9) and 2.9 (range 0.2–83.8), respectively. There were significant increases in post-treatment CRP contrasted with baseline parameter, as post-treatment CRP elevating by median of 131% (range −68% to 6758%) (*p* < 0.001).

Cox regression univariate assessment demonstrated that post-treatment CRP ≥ 2.9 was still significantly linked to inferior OS (HR = 4.20, 95% CI 1.56–11.33, *p* = 0.005) as well as PFS (HR = 4.02, 95% CI 1.46–11.06, *p* = 0.007). However, post-treatment CRP was no longer connected with OS (HR = 1.76, 95% CI 0.48–6.41, *p* = 0.393), but still significantly linked to PFS (HR = 4.83, 95% CI 1.28–18.25, *p* = 0.020), when controlling for the covariates of age, gender, ECOG PS, CCI, tumor pathologic type, T-stage, smoking status, and BED_10_. Our work declared no statistical correlation between post-treatment dNLR and OS (*p* = 0.058), or PFS (*p* = 0.393) (Table 3).

## 4. Discussion

An outstanding dilemma in the SBRT for medically inoperable early-stage NSCLC is selection of patients for the optimal nursing care and use of adjuvant systemic therapy. Although high local cancer management yields, over 70% of SBRT patients suffered regional relapse or distant failure within 5 years, and the 5-year OS is below 50% [4]. Inflammation triggers many molecular processes in cancer cells, which facilitate immune cell evasion and tumorous invasiveness. The immune system performs a crucial part in preventing tumor development and progression [21,22]. Cells of lymphoid and myeloid lineages may have varying physiologic impacts on tumorigenesis, metastasis, and radiation responsiveness, according to preclinical and clinical findings [23,24,25]. B and T lymphocytes may regulate numerous pro-tumorigenic characteristics of myeloid cells throughout radiotherapy [23]. Myeloid cells, such as neutrophils and monocytes, are driven to tumors and increase vasculogenesis, hence negating the treatment’s efficacy and adversely impacting the results of radiation [26,27]. The extent to which this hypothesis is applicable in settings such as SBRT, which high doses over a limited number of fractions, remains to be explored.

A few studies have investigated the association between routinely available peripheral blood biomarkers and prognosis of SBRT, and the pre-treatment NLR, MLR, or PLR are among promising candidates able to predict survival in early-stage NSCLC undergoing SBRT [13,14,15,16]. The results of these studies, however, appear inconsistent, which may attribute to the limited median follow-up time, smaller sample size, and variability in the assessed markers. Our study differs from these analyses in that we comprehensively examined baseline CBC and BBC levels, as well as available post-treatment metrics for subgroup analysis. We demonstrated that high baseline values of dNLR and CRP were associated with worse OS of early-stage NSCLC patients treated with SBRT. Pre-treatment CRP was also independently and strongly associated with PFS, similar result was continued observed in the subsequent subgroup analysis on post-treatment CRP. To our knowledge, this is the first study to explore the role of CRP as potential markers in lung SBRT.

There are several plausible explanations for our findings. One theory is a higher dNLR reflects the gain of a pro-tumorigenic neutrophilia paired with the loss of anti-neoplastic lymphocytes as a component of WBC. Radiotherapy, especially SBRT is known to enhance antigen presentation and release cytokines, that help recruit, activate, and increase lymphocytes [28,29]. Tumors with higher dNLR may possibly be demonstrating a malfunctioning host immunological response may result to incapability to mount an antitumor response following radiation, which correlate with OS. Several recent studies have identified that a greater dNLR or NLR is a prognostic and possibly a predicting indicator for worse therapeutic responsiveness and clinical results, not just following radiotherapy, but also following chemotherapy and immunotherapy. CRP, which is generated as a non-specific acute phase response to so many kinds of inflammation, has been found to be a predictive factor for several tumor forms treated with immune checkpoint inhibitors [30,31,32]. The connection between elevated CRP levels and poor results does not seem to be limited to immunotherapy patients. Forinstance, recently published data by Aires and colleagues found that lower expression of CRP was an independent predictor of improved response to chemoradiotherapy (CRT) and survival for locally advanced rectal cancer patients [33]. Rühle et al. reported that increased CRP level was independent significant parameter for worse OS among 284 neck squamous cell carcinoma patients undergoing CRT [34]. Similar funding reported by Xiao et al. showed that CRP mediated the association with epigenetic age acceleration, patients undergoing radiation therapy for head and neck cancer who had high CRP level exhibited increases of 4.6 years in epigenetic age acceleration compared with those who had low CRP (*p* < 0.001) [35]. Taken together, dNLR and CRP from baseline have prognostic and potentially predictive utility in the setting of cancer.

In this study, deaths from non-tumor causes accounted for about 40% of all deaths, which were mainly attribute to advanced age of the inoperable patients with poor lung function and severe comorbidities. In addition, the long follow-up period that covers the life expectancy of patients is another reason. Baseline dNLR and CRP have also been shown to be independently associated with cancer-specific survival in further analysis, suggesting the specificity of these two parameters in predicting tumor-related causes of death (Appendix A). We also discovered that the pre-treatment dNLR and CRP are mutually reinforcing at predicting OS of patients treated by SBRT. The survival time of patients with none (Group 1) or only one (Group 2) parameter above the cut-off value was significantly longer than that of patients with both dNLR ≥ 1.4 and CRP ≥ 2.9 (Group 3) (Appendix A). As evidence showed synergy between SBRT and immunotherapy in eliminating micrometastastatic disease [36,37], the combination of SBRT and immunotherapy is currently under investigation in full swing. The combined dNLR and CRP parameters that are derived from CBC and BBC could be a potential sensitive prognostic factor, helpful in selecting the subset of early-stage NSCLC patients who may require closer follow-up after SBRT and its combination with systemic therapies such as immunotherapy.

We found a significant decrease in lymphocyte count of peripheral blood after SBRT compared with baseline (*p* < 0.001) (Appendix A), which seems to be inconsistent with the previous findings that radiotherapy activates and increases lymphocytes. The recruitment of lymphocytes to the local tumor area may explain this phenomenon, because we found that patients with decreased lymphocytes after SBRT tended to have a better PFS than patients with flat or elevated lymphocytes (data not shown). In addition, the effect of ionizing radiation on thymus and bone marrow hematopoietic cells is also a possible reason. However, our findings should be interpreted with caution and need confirmation in preclinical researches and larger prospective trials, due to the limited number of subgroup cases.

There are several drawbacks to the present study. Firstly, though multivariate analyses were used to balance all measurable baseline covariates, some unrecorded baseline covariates such as gross tumor volume could be potential confounders. A second point is approximately one-third of patients had no pathologic diagnosis in tumor. We cannot exclude that squamous cell carcinoma may be high proportion in the group with high dNLR and CRP. In view of the fact that most studies, including this study, have confirmed that the prognosis of squamous cell carcinoma undergoing SBRT is significantly worse than that of adenocarcinoma and NSCLC NOS [38,39]. Last but not least, due to the inoperable nature of the patients in our study, several patients had extensive medical co-morbidities, such as rheumatologic illnesses and infections, which can impact systemic inflammatory indicators and OS. Regardless of these obstacles, we conclude that our research has proven the therapeutic significance of dNLR and CRP as simple, widely available, tissue- and cost-free assessments that are strongly linked with SBRT results.

In summary, biomarkers for predicting prognosis of lung cancer patient are necessary in the era of precision medicine. Our work establishes a framework for the noninvasive stratification of inoperable early-stage NSCLC patients treated by SBRT with distinct prognosis, which may assist in selecting optimal nursing care and therapeutic scheme for every individual. Future efforts are needed to longitudinally assess dynamic changes in dNLR and CRP over time in association with treatment response and survival.

## Figures and Tables

**Figure 1 diagnostics-13-00313-f001:**
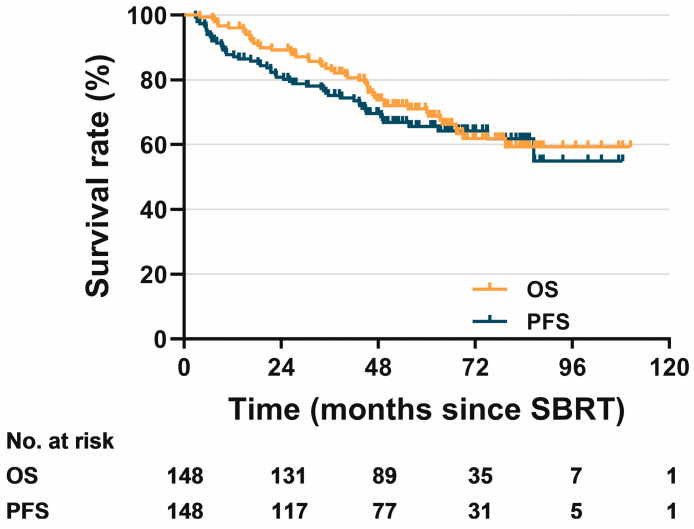
Kaplan–Meier curves for overall survival and progression-free survival of the entire cohort.

**Figure 2 diagnostics-13-00313-f002:**
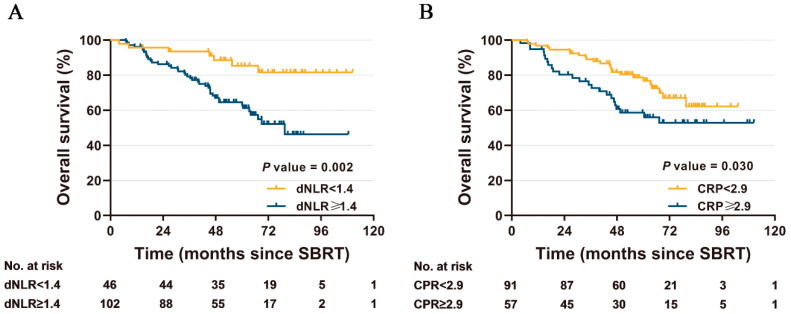
Kaplan-Meier curves for overall survival, stratified by dNLR group (**A**) and CRP group (**B**), respectively.

**Table 1 diagnostics-13-00313-t001:** Demographics, tumor, and treatment characteristics of patients.

Characteristic	*n*	%	Characteristic	*n*	%
**Age (years)**			**T-stage**		
Median (range)	75.5 (47.2–89.0)		T1a	7	4.7
**Gender**			T1b	71	48.0
Female	42	28.4	T1c	51	34.5
Male	106	71.6	T2a	14	9.5
**ECOG PS**			T2b	5	3.4
0	35	23.6	**Location**		
1	86	58.1	Central	5	3.4
2	22	14.9	Peripheral	143	96.6
3	5	3.4	**Site**		
**CCI**			Superior or middle lobe of right lung	40	27.0
Median (range)	3 (2–8)		Inferior lobe of right lung	21	14.2
**Histologic subtype**			Superior lobe of left lung	59	39.9
Adenocarcinoma	48	32.5	Inferior lobe of left lung	28	18.9
Squamous cell carcinoma	27	18.2	**Radiotherapy**		
NSCLC, NOS	21	14.2	50 Gy in 4 or 5 fractions	132	89.2
No pathologic diagnosis	52	35.1	60 Gy in 8 fractions	5	3.4
**Smoking Status**			70 Gy in 10 fractions	8	5.4
Never smoker	59	39.9	Other	3	2.0
Former/current smoker	89	60.1	**BED (Gy)**		
**FEV_1_ Measured (L)**			Median (range)	100 (100–132)	
<1.5	60	40.5	**WBC (×10^9^/L)**		
1.5–2.4	39	26.4	Median (range)	5.7 (2.8–13.3)	
≥2.5	6	4.0	**Neutrophil Count (×10^9^/L)**		
Unknow	43	29.1	Median (range)	3.5 (0.9–9.6)	
**FEV_1_/FVC (%)**			**Lymphocyte Count (×10^9^/L)**		
<70	19	12.8	Median (range)	1.5 (0.6–5.4)	
≥70	86	58.1	**Monocyte count (×10^9^/L)**		
Unknown	43	29.1	Median (range)	0.4 (0.0–1.6)	
**DLCO Measured/Predicted (%)**			**Platelet count (×10^9^/L)**		
<60	27	18.2	Median (range)	194 (73–554)	
60–79	25	16.9	**Serum albumin level (g/L)**		
≥80	57	38.5	Median (range)	42.4 (31.2–67.0)	
Unknow	39	26.4	**LDH (U/L)**		
**PET/CT**			Median (range)	186 (109–354)	
Yes	115	77.7	**CRP (mg/L)**		
No	33	22.3	Median (range)	2.18 (0.02–89.49)	

**Table 2 diagnostics-13-00313-t002:** Cox proportional hazards regression for OS.

Covariables	Univariable Analysis		Multivariable Analysis	
	HR (95% CI)	*p* Value	HR (95% CI)	*p* Value
**Age (year)**	1.03 (0.99–1.06)	0.171	0.98 (0.94–1.03)	0.504
**Gender**				
Female	1 [Reference]		1 [Reference]	
Male	3.52 (1.50–8.28)	**0.004**	4.45 (1.43–13.90)	**0.010**
**ECOG PS**				
0–1	1 [Reference]		1 [Reference]	
2–3	2.97 (1.03–8.57)	**0.044**	5.57 (1.50–20.68)	**0.010**
**Charlson Comorbidity Index**				
2	1 [Reference]		1 [Reference]	
3–4	1.09 (0.60–1.97)	0.784	0.73 (0.39–1.39)	0.343
≥5	1.25 (0.43–3.64)	0.689	0.71 (0.22–2.30)	0.567
**Smoking Status**				
Never smoker	1 [Reference]		1 [Reference]	
Former/current smoker	2.05 (1.07–3.89)	**0.027**	0.78 (0.34–1.80)	0.563
**Pulmonary Function**				
Normal/Mild	1 [Reference]		1 [Reference]	
Moderate	3.12 (0.99–9.83)	0.051	1.05 (0.29–3.80)	0.938
Severe	2.71 (0.92–7.98)	0.070	1.21 (0.37–4.03)	0.753
Unknown	1.90 (0.60–5.95)	0.274	1.09 (0.32–3.74)	0.895
**T-stage**				
T1	1 [Reference]		1 [Reference]	
T2	1.19 (0.67–2.09)	0.551	0.93 (0.49–1.76)	0.822
**Histologic subtype**				
Adenocarcinoma	1 [Reference]		1 [Reference]	
Squamous cell carcinoma	2.53 (1.13–5.67)	**0.024**	1.24 (0.50–3.04)	0.646
NSCLC, NOS	1.28 (0.47–3.47)	0.626	0.71 (0.23–2.21)	0.549
No pathologic diagnosis	1.65 (0.78–3.50)	0.190	1.85 (0.82–4.15)	0.138
**BED (Gy)**	0.95 (0.90–1.00)	**0.033**	0.92 (0.87–0.97)	**0.004**
**dNLR**				
<1.4	1 [Reference]		1 [Reference]	
≥1.4	3.38 (1.51–7.58)	**0.003**	4.62 (1.89–11.27)	**0.001**
**MLR**				
<0.3	1 [Reference]			
≥0.3	1.63 (0.88–2.99)	0.119		
**PLR**				
<128	1 [Reference]			
≥128	1.05 (0.60–1.85)	0.868		
**Serum albumin level (g/L)**				
<43.0	1 [Reference]		1 [Reference]	
≥43.0	0.40 (0.21–0.77)	**0.006**	0.57 (0.27–1.22)	0.148
**LDH (U/L)**				
<240	1 [Reference]			
≥240	1.03 (0.37–2.87)	0.954		
**CRP**				
<2.9	1 [Reference]		1 [Reference]	
≥2.9	1.85 (1.05–3.27)	**0.033**	2.92 (1.49–5.70)	**0.002**

The bolded number indicates a statistically significant.

**Table 3 diagnostics-13-00313-t003:** Post-treatment peripheral blood biomarkers and Survival and Disease Control.

Covariables	Univariable Analysis		Multivariable Analysis	
	HR (95% CI)	*p* Value	HR (95% CI)	*p* Value
**OS**				
dNLR	1.46 (0.43–4.91)	0.546	0.71 (0.17–2.94)	0.640
Serum albumin level	0.34 (0.08–1.43)	0.140	0.11 (0.02–0.81)	**0.030**
CRP	4.20 (1.56–11.33)	**0.005**	1.76 (0.48–6.41)	0.393
**PFS**				
dNLR	1.06 (0.36–3.16)	0.915	0.43 (0.11–1.72)	0.232
Serum albumin level	0.44 (0.10–1.90)	0.272	0.27 (0.05–1.57)	0.146
CRP	4.02 (1.46–11.06)	**0.007**	4.83 (1.28–18.25)	**0.020**
**LRR**				
dNLR	0.45 (0.11–1.80)	0.257	0.14 (0.02–1.22)	0.075
Serum albumin level	0.48 (0.06–3.87)	0.493	0.24 (0.02–2.54)	0.238
CRP	4.29 (0.89–20.70)	0.070	5.99 (0.94–38.19)	0.058
**DM**				
dNLR	0.89 (0.30–2.70)	0.842	0.45 (0.10–2.07)	0.305
Serum albumin level	0.22 (0.03–1.66)	0.143	0.11 (0.01–1.54)	0.078
CRP	4.73 (1.56–14.38)	**0.006**	5.56 (1.39–22.21)	**0.015**

The bolded number indicates a statistically significant.

## Data Availability

Data can be made available upon reasonable request.

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
