# Peer review of "Derived Neutrophil-Lymphocyte Ratio and C-Reactive Protein as Prognostic Factors for Early-Stage Non-Small Cell Lung Cancer Treated with Stereotactic Body Radiation Therapy"

_diagnostics, 2023, doi:10.3390/diagnostics13020313_

Round 1

Reviewer 1 Report

Dear authors 

Greetings. The well-written manuscript regarding the association between laboratorial findings and NSCLC prognosis in early-stage cancer is not only of interest at present but also would be a high-yield point in the future of clinical practice. I kindly appreciate your efforts. 

Author Response

Manuscript ID: diagnostics-2114750

Article Title: Derived neutrophil-lymphocyte ratio and C-reactive protein as prognostic factors for early-stage non-small cell lung cancer treated with stereotactic body radiation therapy

Dear Dr. Sabrina Shen and Reviewers:

On behalf of all the authors, I would like to thank you and the reviewers for your response and comments regarding our manuscript. We feel privileged and lucky that our manuscript went to these experts as the valuable comments from them not only enabled us with the improvement of the manuscript, but suggested some neat ideas for future studies.

After serious discussion of these comments from two reviewers by all authors, we have carefully revised the original manuscript to make our results convincing. We also carefully proof-read the manuscript to minimize grammatical and typographical errors. Here, we resubmitted the R1 version of the manuscript. All changes we have made were tracked by Microsoft Word and marked in RED in the manuscript.

We hope that the modifications we have made resolve all your concerns about the article. At the same time, we are also more than happy to make any further corrections to improve the paper and facilitate successful publication. Should you have any questions, please connect us without hesitation.

Thank you and all the reviewers again for the kind advice. I look forward to hearing from you!

Sincerely,

Dr. Baiqiang Dong, on behalf of all authors.

Responses to the comments of reviewers

Reviewer #1:

The well-written manuscript regarding the association between laboratorial findings and NSCLC prognosis in early-stage cancer is not only of interest at present but also would be a high-yield point in the future of clinical practice. I kindly appreciate your efforts.

Response: We appreciate the reviewer for the high opinion of our manuscript.

Reviewer 2 Report

Major comments:

1.     Please also show PFS in dNLR and CRP in Figure 2 if you have the data.

2.     How about OS and PFS data by combined NLR and CRP in Figure 2. Is it better or not.

3.     Did you investigate dNLR and CRP in patients without SBRT, and their relationship with OS and PFS as a control group.

4.     If NLR is associated with tumor progression and poor survival rate, SBRT decrease it that is good for patients. But why CRP also associate with tumor progression and poor survival rate but that is increased in SBRT treatment? Do you have any explanation for the observation. 

Minor comments:

1.     Please revise “Methords” in section 2 (Line 71).

Author Response

Manuscript ID: diagnostics-2114750

Article Title: Derived neutrophil-lymphocyte ratio and C-reactive protein as prognostic factors for early-stage non-small cell lung cancer treated with stereotactic body radiation therapy

Dear Dr. Sabrina Shen and Reviewers:

On behalf of all the authors, I would like to thank you and the reviewers for your response and comments regarding our manuscript. We feel privileged and lucky that our manuscript went to these experts as the valuable comments from them not only enabled us with the improvement of the manuscript, but suggested some neat ideas for future studies.

After serious discussion of these comments from two reviewers by all authors, we have carefully revised the original manuscript to make our results convincing. We also carefully proof-read the manuscript to minimize grammatical and typographical errors. Here, we resubmitted the R1 version of the manuscript. All changes we have made were tracked by Microsoft Word and marked in RED in the manuscript.

We hope that the modifications we have made resolve all your concerns about the article. At the same time, we are also more than happy to make any further corrections to improve the paper and facilitate successful publication. Should you have any questions, please connect us without hesitation.

Thank you and all the reviewers again for the kind advice. I look forward to hearing from you!

Sincerely,

Dr. Baiqiang Dong, on behalf of all authors.

Reviewer #2:

1. Please also show PFS in dNLR and CRP in Figure 2 if you have the data.

Response: We thank the reviewer for the constructive suggestion. In this study, the multivariable analysis showed that dNLR ≥ 1.4 and CRP ≥ 2.9 were statistically and independently linked to worse overall survival, and the CRP ≥ 2.9 remained independent significant parameters for inferior PFS (HR = 3.13; 95% CI 1.63–6.01; P = 0.001) However, there was no longer statistically significant correlation between dNLR and PFS (P = 0.100). Due to space constraints, the detailed instruction of PFS in dNLR and CRP was recorded in the supplementary materials (please see manuscript Page 5, line 181-186, Table S3, and Figure S3B).

2. How about OS and PFS data by combined NLR and CRP in Figure 2. Is it better or not.

Response: Thank you for your question. In this study, we discovered that the pre-treatment dNLR and CRP are mutually reinforcing at predicting OS of patients treated by SBRT. The survival time of patients with none (Group 1) or only one (Group 2) parameter above the cut-off value was significantly longer than that of patients with both dNLR ≥ 1.4 and CRP ≥ 2.9 (Group 3) We have added supplementary introduction to this aspect in the DUSCISSION section (please see Page 9, line 266-270). However, as stated above, there was no statistically significant correlation between dNLR and PFS, thus we did not explore the role of dNLR combined with CRP in PFS.

3. Did you investigate dNLR and CRP in patients without SBRT, and their relationship with OS and PFS as a control group.

Response: We thank the reviewer for the constructive comments. SBRT has been observed to yield greater antitumor effectiveness than would be predicted from standard radiobiologic modeling alone, possibly through superior engagement of the immune system, leading to enhanced antitumor immunity1-3. Therefore, it will be interesting to compare dNLR and CRP levels in lung cancer patients with SBRT versus conventional radiotherapy or surgery. However, due to the limitation of the sample, we only analyzed the SBRT cohort in the present study. Future work requires a lateral assessment of dNLR and CRP in relation to treatment patterns.

4. If NLR is associated with tumor progression and poor survival rate, SBRT decrease it that is good for patients. But why CRP also associate with tumor progression and poor survival rate but that is increased in SBRT treatment? Do you have any explanation for the observation.

Response: Thank you for your question. In this study, there were significant increases in post-treatment CRP contrasted with baseline parameter, as post-treatment CRP elevating by median of 131% (P < 0.001), which similar with the previous reports (please see Page 8, line 250-254)4-6. A plausible explanation for this finding is that apoptosis and necrosis of tumor cells occurred after SBRT, which led to an increase in CRP levels. However, higher inflammation induces several molecular cascades in cancer cells, and promoting tumor invasion and immune cell escape7. However, the significance of this increase in CRP after SBRT is still unclear, this may represent a sustained posttreatment inflammatory state induced by radiotherapy.

5. Please revise “Methords” in section 2 (Line 71).

Response: Thank the expert for the careful reading, and we have made revision according to the reviewer’s comments (please see Page 2, line 71).

References

  1. Zhou P, Chen D, Zhu B, et al. Stereotactic Body Radiotherapy Is Effective in Modifying the Tumor Genome and Tumor Immune Microenvironment in Non-Small Cell Lung Cancer or Lung Metastatic Carcinoma. Front Immunol. 2021;11:594212.
  2. Liu C, Sun B, Hu X, et al. Stereotactic Ablative Radiation Therapy for Pulmonary Recurrence-Based Oligometastatic Non-Small Cell Lung Cancer: Survival and Prognostic Value of Regulatory T Cells. Int J Radiat Oncol Biol Phys. 2019;105(5):1055-1064.
  3. Brown JM, Carlson DJ, Brenner DJ. The tumor radiobiology of SRS and SBRT: are more than the 5 Rs involved? Int J Radiat Oncol Biol Phys. 2014;88(2):254-62.
  4. Aires F, Rodrigues D, Lamas MP, et al. C-Reactive Protein as Predictive Biomarker for Response to Chemoradiotherapy in Patients with Locally Advanced Rectal Cancer: A Retrospective Study. Cancers (Basel). 2022;14(3):491.
  5. Rühle A, Stromberger C, Haehl E, et al. Development and validation of a novel prognostic score for elderly head-and-neck cancer patients undergoing radiotherapy or chemoradiation. Radiother Oncol. 2021;154:276-282.
  6. Xiao C, Beitler JJ, Peng G, et al. Epigenetic age acceleration, fatigue, and inflammation in patients undergoing radiation therapy for head and neck cancer: A longitudinal study. 2021;127(18):3361-3371.
  7. Douglas H, Robert A. Hallmarks of cancer: the next generation. 2011;144(5):646-74.

Round 2

Reviewer 2 Report

1.     Please revise “Methords” in section 2 (Line 71).